# Mitochondrial DNA Variation in Peruvian Honey Bee (*Apis mellifera* L.) Populations Using the tRNA^leu^-cox2 Intergenic Region

**DOI:** 10.3390/insects12070641

**Published:** 2021-07-14

**Authors:** Julio Chávez-Galarza, Ruth López-Montañez, Alejandra Jiménez, Rubén Ferro-Mauricio, Juan Oré, Sergio Medina, Reyna Rea, Héctor Vásquez

**Affiliations:** Dirección de Desarrollo Tecnológico Agrario, Instituto Nacional de Innovación Agraria, Av. La Molina 1981, Lima 15024, Peru; ruthlopezmont50@gmail.com (R.L.-M.); ajimenez.pnia124@gmail.com (A.J.); rubenferro20@gmail.com (R.F.-M.); juancarlosore2507@gmail.com (J.O.); sj.med.agro@gmail.com (S.M.); reynareazenozain@gmail.com (R.R.); hvasquez@inia.gob.pe (H.V.)

**Keywords:** *Apis mellifera*, haplotype, mitochondrial diversity

## Abstract

**Simple Summary:**

Currently, the genetic diversity of Peruvian honey bee populations is unknown. Only two studies were carried out and suggest that many regions of Peru present Africanized honey bee colonies and a varied degree of Africanization. To molecularly characterize and know more about the genetic background of Peruvian honey bees, the highly polymorphic tRNAleu-cox2 was used. This study analyzed 512 colonies in three regions of Peru: Lima, Piura, and Junín. The results indicated that 65% colonies correspond to lineage A (African), 33.8% colonies to lineage C (Eastern European), and 1.2% colonies to lineage M (Western European). A total of 24 haplotypes were identified: 16 haplotypes belong to lineage A (sub-lineage A_I_ (13), sub-lineage A_III_ (03)), lineage C (06), and lineage M (02), and 15 of them are for the first time reported and represented by A1t, A1u, A1w, A4p, A4q, A4s, A4t, A4u, A4v, A4w, 30d, A30e, A65, M7b, and M7c. Piura and Lima presented higher proportions of African haplotypes and lower proportions of haplotypes from lineage C than Lima. Very few haplotypes of lineage M were identified, whose presence could be due to accidental purchases or traces of honey bee introductions from lineage M in the 19th century. Hence, studies about the diversity and genetic structure of Peruvian honey bee populations are necessary to promote adequate, sustainable management and establish conservation and breeding programs.

**Abstract:**

Mitochondrial DNA variations of Peruvian honey bee populations were surveyed by using the tRNA^leu^-cox2 intergenic region. Only two studies have characterized these populations, indicating the presence of Africanized honey bee colonies in different regions of Peru and varied levels of Africanization, but the current status of its genetic diversity is unknown. A total of 512 honey bee colonies were sampled from three regions to characterize them. Our results revealed the presence of European and African haplotypes: the African haplotypes identified belong to sub-lineage A_I_ (13) and sub-lineage A_III_ (03), and the European haplotypes to lineages C (06) and M (02). Of 24 haplotypes identified, 15 new sequences are reported here (11 sub-lineage A_I_, 2 sub-lineage A_III_, and 2 lineage M). Peruvian honey bee populations presented a higher proportion from African than European haplotypes. High proportions of African haplotype were reported for Piura and Junín, unlike Lima, which showed more European haplotypes from lineage C. Few colonies belonging to lineage M would represent accidental purchase or traces of the introduction into Peru in the 19th century.

## 1. Introduction

The honey bee (*Apis mellifera* L.) plays an important pollinator role worldwide, and is considered the most frequent floral visitor of agriculture crops and natural habitats [1]. Its contribution to the global economy for food production is estimated between USD 235 and 285 billion annually [2]. Additionally, honey bees provide other benefits with a strong impact on ecosystem conservation and preservation, as well as human nutrition and health [3].

The natural distribution of *A. mellifera* encompasses Europe, Africa, the Middle East, and parts of Asia [4,5]. Within this geographical area, the dispersion capacity of *A. mellifera* to colonize the several habitable ecosystems and adapt to different bioclimatic conditions has led to the emergence of over 33 subspecies [6]. Based on morphological studies, these subspecies were initially classified into four major evolutionary lineages: African (A), Western and Northern European (M, hereafter referred to as Western European), Eastern European (C), and Western and Central Asian (O) [4]. In recent years, two new Middle East lineages have been proposed, named Y and S, using molecular markers [7,8]. In addition, the African lineage has been divided into four sub-lineages (A_I_, A_II_, A_III_, and Z) [7,9].

Numerous studies have been performed to explain and understand the evolutionary history and pattern of genetic diversity of honey bee populations. These studies have used several markers, including morphology [4,10,11], allozymes [10,12,13,14], mitochondrial DNA [7,8,9,15,16,17,18], microsatellites [7,16,19,20], and SNPs [21,22,23]. Among all the markers, mitochondrial DNA variation in the highly polymorphic tRNA^leu^-cox2 intergenic region is the molecular marker most widely used to distinguish lineages, sub-lineages, and genetic structure in *A. mellifera* [24].

The tRNA^leu^-cox2 intergenic region is characterized by the presence of a non-coding sequence formed by two elements (P and Q) and part of the cox2 region. The P element varies in size (~53–68 bp) and shows the four forms (P_0_, P, P_1_, P_2_), allowing the distinction of evolutionary lineages and sub-lineages. The Q element varies between ~194 and 196 bp, which can be repeated in tandem one to five times, although the number of repeats is not lineage-specific. The A lineage is characterized by the P_0_ (no large deletion) and P_1_ form (15 bp deletion at the 3′ end), the M lineage by the P form (13 bp deletion in the middle of P element), the Y lineage by the P_2_ form (18 bp deletion at the 5′ end of P element), and the C lineage lacks the P element [7]. Regarding the distinction of African sub-lineages, these sub-lineages are commonly identified by the presence and absence of *Dra*I site (TTTAAA) and P_0_ or P_1_ forms [7,9]. Sub-lineages A_I_, A_II_, and Z present the P_0_ form, and sub-lineage A_III_ carries the P_1_ form. The differences among sub-lineages A_I_, A_II_, and Z lie in the *Dra*I site, sub-lineage AI with *Dra*I sites at the 3′ end of the tRNA^leu^ and the 5′ end of the first Q element, sub-lineage A_II_ with *Dra*I site at the 3′ end of the tRNA^leu^ only, and sub-lineage Z with *Dra*I sites at the 3′ end of tRNA^leu^, the 5′ end, and in the middle of the first Q element. This complexity and variation makes the tRNA^leu^-cox2 intergenic region suitable for detecting and tracking importation of queens, phylogeographical studies, conservation strategies, and for understanding processes as introgression or contact among populations with different origins [17,18,25,26].

In the Americas, the first European honey bee populations were introduced by settlers from England in 1617 [27]. The arrival in South America of the first beehives occurred in Brazil [28], and then they were introduced to Chile and Peru in 1884 [14,28]. In 1974, the presence of Africanized honey bees was reported for the first time in the Peruvian eastern tropical area of Pucallpa [29]. Gradually, the dispersion of Africanized honey bee swarms across Peruvian territory has displaced resident European honey bees in their advance. To date, little information is available on the diversity and genetic structure of Peruvian honey bee populations. A morphometric study concluded that Africanized honey bee colonies were found in all Peruvian regions under study, with the highest predominance in the tropical region, producing varied levels of Africanization due to hybridization between Africanized and European honey bees [30].This was further supported by integrating RFLP-mtDNA and morphometrics data where a process of introgressive hybridization between Africanized and European honey bees would better explain the pattern of distribution across the different regions of Peru [31]. However, there are gaps regarding evolutionary lineages or sub-lineages that constitute the current genetic structure in Peruvian populations. Therefore, the aim of this study is to analyze the status of mitochondrial DNA variations in honey bee populations from three regions of Peru, based on sequence data of the tRNA^leu^-cox2 intergenic region. Although many studies estimating mitochondrial diversity of honey bees have obtained it using restriction digestion data from the whole mitochondrial molecule [15,32] or from the tRNA^leu^-cox2 region [7,9,16,17,18,33,34,35], this is the first study reporting on maternal diversity patterns using sequence data for Peruvian honey bee populations.

## 2. Materials and Methods

### 2.1. Sampling Sites

Between 2017 and 2018, a sampling was carried out in three Peruvian regions: Lima, Piura, and Junín. Five sampling sites were considered for each region (Appendix A). For each sampling sites, apiaries were sampled within a 15 km radius circle, resulting in three apiaries per site, with exception for La Matanza 1 (one apiary). A total of 512 colonies were sampled in 43 apiaries and grouped into 15 sampling sites. Workers were collected from the inner part of hives, placed in absolute ethanol, and stored at −20 °C until molecular analysis. The geographical locations of each sampling sites are shown in Figure 1.

### 2.2. Mitochondrial DNA Analysis

Total DNA was extracted from the thorax of the 512 individuals, each representing a single colony using the phenol-chloroform isoamyl alcohol (25:24:1) protocol [36]. Analysis of mtDNA was based on the tRNA^leu^-cox2 mitochondrial intergenic region. This region was PCR-amplified using the primer pair E2 (5′-GGCAGAATAAGTGCATTG-3′) and H2 (5′-CAATATCATTGATGACC-3′), according to a protocol detailed in Garnery et al. [37]. The amplified PCR products were sent to Macrogen (Seoul, Korea) for direct sequencing in both directions with primers E2 and H2. The sequences were checked for base calling and aligned with published sequences available in GenBank (http://www.ncbi.nlm.nih.gov/genbank, accessed on 17 December 2020) using MEGA X [38]. The new haplotypes and variants were determined following the criteria of Chávez-Galarza et al. [17] and Rortais et al. [35].

### 2.3. Genetic Diversity and Phylogenetic Analysis

The GENEALEX 6.5 program [39] was used to obtain the following genetic diversity parameters within each sampling site: mean number of alleles per locus (Na), effective number of alleles (Ne), number of private alleles (Np), and unbiased diversity (uh). Genetic differentiation among sampling sites based on lineages and sub-lineages was estimated using Φ*_PT_* values, which were then employed to perform a principal coordinate analysis (PCoA) with the same program.

Phylogenetic analysis of haplotypes was performed using the PHYLIP package 3.65c [40]. The phylogenetic tree allowed the identification of the lineages and sub-lineages to which the novel haplotypes belong. Reference haplotypes from lineages A, M, C, and Y obtained from NCBI (http://www.ncbi.nlm.nih.gov/, accessed on 17 December 2020) were included in the analysis [7,9,17,41,42,43]. The analyses included polymorphic sites of the P element, the first Q element, and the fragment of cox2. The second Q element, present in some haplotypes, was considered single mutational steps. Gaps were considered as a fifth character. Positions indicating large deletions within the P element (P, P_1_, P_2_, and absence of P) and the first two restriction sites, based on the *Dra*I test [37], were upweighted. The tRNA^leu^-cox2 region of *Apis cerana* was used as outgroup. The absence of the P element in lineage C and the absence of the Q element in *A. cerana* were coded as a single mutational step and the remaining positions as missing data.

Proximity among haplotypes and their proportions in the three Peruvian regions were investigated using the median-joining network algorithm [44] as implemented in the program Network version 4.6.1.1 (Fluxus Engineering, Clare, UK; http://www.fluxus-engineering.com, accessed on 7 August 2013). The polymorphic sites within the P and Q elements were used to infer the network. Deletions above 2 bp were considered as a single-mutational step and thus coded as a 1 bp gap. As for phylogenetic analysis, positions indicating large deletions within the P element (P, P_1_ and absence of P) and the first two restriction sites were also upweighted.

## 3. Results

Based on the tRNAleu-cox2 intergenic region, haplotypes representing three different evolutionary lineages were detected among Peruvian honey bee samples: African (A), Western European (M), and Eastern European (C). Six different length patterns, resulting from combinations of the different P and Q elements, were observed among the 512 samples analyzed. For the first time, a P_0_ pattern is reported (three samples), along with well-known patterns such as Q (173 samples), P_0_Q (55 samples), P_0_QQ (222 samples), PQQ (six samples), and P_1_QQ (53 samples).

### 3.1. New Haplotype and Variants

Sequencing of the tRNA^leu^-cox2 intergenic region generated from 512 honey bee individuals revealed 333 haplotypes belonging to lineage A, 173 to lineage C, and 6 to lineage M. Within lineage A, 280 individuals correspond to sub-lineage A_I_ and 53 to sub-lineage A_III_. The analysis of this region reported 24 haplotypes (accession numbers: MW677198–MW677221), 16 correspond to lineage A (13 sub-lineage A_I_ and 3 sub-lineage A_III_), 6 to lineage C, and 2 to lineage M. Among these haplotypes, 15 are reported here for the first time (11 sub-lineage A_I_, 2 sub-lineage A_III_, and 2 lineage M).

Characteristics of novel haplotype and variants are shown in Figure 2. For lineage A, one new band pattern was named A65 with a length of 422 bp, whose main characteristic is the absence of the Q element. New variants of haplotype A1 were reported and denominated A1t, A1u, and A1w. Variants A1t and A1u shared substitution G/A in position 99, and A1t with A1w the substitution C/T in position 646. The difference among them is due to an insertion A/– in position 139 for A1t, a substitution T/C in position 286 for A1u, and an insertion G/– in position 100 for A1w. With respect to haplotype A4, the variants A4p and A4q displayed a substitution A/G in position 99, an insertion G/– in position 100, and substitution C/T in position 646, the unique difference among both was an insertion A/– in position 298. The variant A4s presented an insertion A/– in position 337 and substitution C/T in position 646. The variants A4t, A4u, A4v, and A4w shared a deletion –/T in position 252 and substitution C/T in position 646. The insertion A/– in position 337 was in A4t, A4v, and A4w. The difference among A4v and A4w is a substitution A/T in position 739 and A/T in position 795, respectively. In our study, two new variants of haplotype A30 presenting a deletion –/C in position 336 for A30d and substitution C/A in position 86 for A30d are reported. Within lineage M, new variants M7b and M7c were identified for haplotype M7. Both of them shared a substitution A/G in position 84, and only M7c presented a further substitution T/C in position 148.

### 3.2. Phylogenetics Analysis of New Haplotypes and Variants

Phylogenetic analysis confirmed the novelty of 15 haplotypes of African and Western Europe ancestry (Figure 3). These novel haplotypes were grouped within lineages A (13) and M (02). In lineage A, 11 novel haplotypes belong to sub-lineage A_I_, 2 to sub-lineage A_III_, and none were placed within sub-lineage A_II_ nor Z. Haplotypes already reported and also identified in this study (A1, A1e, A30, C1, C2, C2c, C2l, C2j, C3) were grouped in their respective lineages, as previously reported [17]. The phylogeny supported three African sub-lineages (A_I_, A_II_, A_III_) forming a cluster, although A_I_ is not differentiated. Sub-lineage Z forms a group well-differentiated but separated from other African sub-lineages, and lineage Y was not well-differentiated. Haplotypes of M and C ancestry formed a well-supported cluster.

### 3.3. Distribution of Haplotypes

Lineage A (333 colonies, 65.0%) was dominant as compared with lineages C (173 colonies, 33.8%) and M (six colonies, 1.2%). Within lineage A, the sub-lineage A_I_ (280 colonies, 84.1%) was also more ubiquitous than sub-lineage A_III_ (53 colonies, 15.9%). Of the 24 different haplotypes identified in Peruvian honey bee populations (Figure 1), the haplotypes identified with the highest frequency were A4p (117 colonies, 22.8%), A4t (96 colonies, 18.8%), A30 (51 colonies, 9.9%), C1 (94 colonies, 18.4%), and C2j (60 colonies, 11.7%), and haplotypes with the lowest frequency were A1t, A1w, A4q, A4s, A4u, A4v, A4w, A30d, A30e, A65, M7b, M7c, C2, C2l, and C3, with a percentage below 0.9%. The most distributed haplotypes are A4p and A4t, with presence in 13 and 11 sampling sites within three Peruvian regions, respectively. On the other hand, some haplotypes were exclusively present in one sampling site such as A1t, A30d, A30e, and C3 in Region Lima; A4q, M7c, and C2l in Region Piura; and A1w, A4s, A4v, and A4w in Region Junín (see Appendix A).

### 3.4. Diversity Measures

Diversity measures estimated for each sampling sites are shown in Table 1. The number of haplotypes (Na) ranged from three (Barranca and Coayllo) to nine (Quilmaná). Sampling sites La Matanza 1 and San Ramón presented the highest values of effective number of alleles (Ne). For the number of private alleles (Np), all sampling sites showed one. The unbiased haplotype diversity (uh) ranged from 0.181 (Barranca) to 0.879 (La Matanza 1). In general, the high values of diversity were found in all sampling sites from Piura and Junín Regions as compared to Lima.

### 3.5. Relationships among Haplotypes and Lineages

A median-joining network that illustrates the frequencies and relationships among the haplotypes found in the Peruvian honey bee populations are shown in Figure 4. Three highly divergent clusters were differentiated, which correspond to lineages A, C, and M. For lineage A, the two African sub-lineages were mainly linked by haplotypes A4p, A4t, A1e (sub-lineage A_I_), and A30 (sub-lineage A_III_). The majority of A4 variants identified correspond to Junín and diverged from A4t, which is also located in the same region with the highest frequency. However, haplotype A4p is the most distributed only one variant A4 (A4q) diverged from it, and both of them were identified in Piura. Variants A30d and A30e diverged from A30, and are only present in Lima. Although A1e displays more links, only A1u would have directly diverged from it and the remains are hypothetical haplotypes. The new haplotype A65 presented most divergence within lineage A is due to the lack of Q elements, deletions/insertions, and substitutions. For lineage C, variants C3 and C2j diverged from C2, whereas C2l and C1 diverged from C2c, presenting all of them with more frequency in Lima region. Few colonies presented variants of haplotype M7 (M7b and M7c), and only one position produced divergence between them. In the analysis of PCoA based on lineages and African sub-lineages (Figure 5), the main axis explained 70.2% of the genetic variation separating sampling sites of Lima with a higher frequency of haplotypes belonging to lineage C than the sampling sites of Piura and Junín that present more haplotypes of lineage A. The second axis explained 15.5% of the genetic variation and showed differentiation between sampling sites with haplotypes belonging to sub-lineage A_III_ and sampling sites with more frequency of haplotypes belonging to sub-lineage A_I_ and lineage C.

## 4. Discussion

In this study, we reported for the first time sequences of haplotypes based on tRNA^leu^-cox2 sequences from Peruvian honey bee populations. Our study revealed the presence of nine haplotypes described previously as A1, A1e, A30, C1, C2, C2c, C2j, C2l, C3, one new haplotype A65, and 14 new variants of haplotypes A1 (A1t, A1u, A1w), A4 (A4p, A4q, A4s, A4t, A4u, A4v, A4w), A30 (A30d, A30e), and M7 (M7b, M7c). The most frequent African haplotypic patterns identified were A1, A4, and A30. Within African sub-lineage A_I_, the A4 pattern (43.4%) presented a higher proportion than the A1 pattern (10.7%); similar results were reported for Brazil, Uruguay, and Venezuela [41,45,46], although Mexico, Colombia, Argentina, Costa Rica, and the USA showed a predominance of A1 with respect to A4 [7,45,47,48,49,50]. Meanwhile, the A30 pattern, belonging to sub-lineage A_III_, was the third most frequent (10.4%), coinciding with the results reported for Brazil and Uruguay [41,46], but contrasting with the low frequency (≤0.5%) found for Colombia and the USA [49,50]. For lineage C, our results showed haplotypic patterns C1 (18.2%) and C2 (14.8%) as the most frequent. The presence of C1 and C2 were also reported in colonies from Uruguay, the USA, and Chile [46,51,52], but only C1 was found in colonies from Argentina, Colombia, Venezuela, Mexico, and Brazil [45,48,50]. A unique haplotypic pattern M7 (1.2%) was found in Peruvian colonies; the same pattern was reported in colonies from Brazil, Uruguay, Mexico, and Chile [41,45,46,52].

Regarding to the presence of haplotypic pattern A4, this pattern has been previously reported in Iberia and Africa [7,17]. Sequences of their variants were reported for Brazil and Uruguay, and have also been observed in samples from South Africa and Tanzania [18]. This agrees with an African origin of variants A4. Its arrival was first in Brazil and then to other close countries [53]. However, Peruvian colonies presented new variants different from those reported in Brazil. Haplotype A1 and its variant A1e were previously reported from the Iberian Peninsula and the USA, respectively [17,49], and we found them mainly distributed in Junín and Piura. Since the date of introduction of Africanized honey bee into Peru is known [54], it seems most likely that these haplotypes were introduced from Iberian colonies as Iberian honey bee populations possess African and Western European haplotypes because of its hybrid origin between lineages M and A [23,55]. These haplotypes would have entered via Portuguese settlers to Brazil and later expanded during the spread of Africanized honey bee. This hypothesis is reinforced by the presence of haplotype A30, belonging to sub-lineage A_III_, in Peruvian colonies, whose high frequency was reported in Northern Portugal [17,34]. Therefore, African haplotypic patterns present in Peruvian colonies probably have African and European origins. Representative presence of haplotypes of lineage C was reported for honey bee populations from Lima; this result is not surprising as many beekeepers around the world have preference for Italian honey bees due to their easy handling [4]. Moreover, the importations of queens into Peru come from Chile, whose colonies mainly descend from honeybees belonging to lineage C (*Apis mellifera ligustica* and *A. mellifera carnica*) and some of them to lineage M (*Apis mellifera mellifera*) [52]. A very low frequency of haplotypic pattern M7 could be due to accidental purchases of M ancestry queens or traces of honey bees that arrived from Iberian and the Italian Peninsula or France [17,56,57] at the beginning of introductions of honey bees into Peru [27].

Our results found more haplotypes of lineage A in Junín and Piura than in Lima. The distribution of these haplotypes agrees with the migratory route and occupation range of Africanized honey bees proposed by Kent [54]. Swarms of Africanized honey bees had arrived in the region of Ucayali in 1974 then appeared in the neighboring regions of Junín and Pasco. This expansion stretched until the Andean foothills because of the geographic barrier of the Andes Mountains. However, the presence of African haplotypes on the Pacific coast may be explained by the introduction of Africanized colonies that came from tropical rainforests located on the eastern side of the Andes, via the beehives trade [54]. This is demonstrated by a similar presence and high proportions of haplotypic patterns A1 and A4 reported for Piura (northern coast) and Junín (Amazonian region). The few A1, A4, and A30 patterns found in Lima were initially the result of natural spread of Africanized swarms from north to south and then a constant replacement by imported queens of lineage C. The importation or selection of queens, descendants of lineage C, is an activity widely practiced by beekeepers in Lima, which is corroborated in this study by the high frequency of haplotypic patterns C1 and C2 found. Beehives or queens purchases from Lima or other countries still impact the presence of haplotypes of lineages C and M in other regions of Peru, as showed for Junín and Piura.

We reported for the first time haplotype A65 exhibiting a striking architecture presenting the P element only (Figure 2), as reported for *Apis cerana* (the closest relative of *A. mellifera*) [58]. The phylogenetic analysis suggests that this haplotype corresponds to African sub-lineage A_I_. Its presence was reported in three colonies (two in Piura and one in Junín), and no studies to date performed with samples belongings to areas of natural distribution for honey bees have found this haplotypic pattern. Regarding to new variants, the number found was unexpected due to the historical introductions of honey bees to Peru, since these come from European honey bees and more recently from Africanized honey bees. Tibatá et al. [50] reported 18 haplotypes using the same intergenic region for Colombian honey bee populations, but all of them were previously published. This lack of new haplotypes in Colombian honey bee populations might be due to the practice of intensive professional beekeeping, as beekeepers have had better control and selection of their honey bee colonies since the arrival of Africanized honey bee to Colombia caused a kind of bottleneck, thus losing part of the mitochondrial genetic diversity that would involve less frequent haplotypes. This situation does not occur in many regions of Peru, where amateur or hobbyist beekeeping is mainly practiced in areas with Africanized honey bee populations. Many Peruvian beekeepers face the loss of beehives by replacing them with feral honey bee colonies from an unknown mitochondrial genetic background, which would explain the new haplotypes found and not reported in other studies. In contrast, professional beekeepers, located mainly in Lima, prefer to buy and maintain nuclei or queens with a well-established lineage C.

On the other hand, climatic and environmental factors present in Peruvian territory favored a better fitness and adaptability of Africanized honey bees over European honey bees, as reported in several countries of the American continent [46,50,59,60]. These factors create, in turn, a selection pressure on honey bee genomes, and mitochondrial DNA is no exception. Evidence of selection pressures on mitochondria has been reported [61], and its effect on individuals could be associated both with the fixation of certain haplotypes and the formation of new ones. This might also explain the presence of new haplotypes and variants, and the presence of A30 in Piura and Lima. Curiously, A30 should have been found in Junín as this region was one of the first to be colonized by Africanized honey bees from Brazil during its expansion into Peruvian territory, but its presence was onlyreported in the coastal zone. This particularity was also reported for Northern Portugal, a region near to the Atlantic side with the highest diversity of haplotypes belonging to sub-lineage A_III_ [17]. Although we do not have empirical support or indirect evidence from sequence variation analyses, we suggest that selection might have had a major role in shaping the observed mitochondrial diversity in Peru, a hypothesis that certainly deserves further studies.

The current genetic diversity of Peruvian honey bee populations found in this study is the result of a short period of ecological adaptation to the environment and interaction between different honey bee subspecies such as *A. mellifera*, *A. m. iberiensis*, *A. m. ligustica*, *A. m. carnica*, and *A.m. escutellata*. Taking all these together, the Peruvian territory could still harbor important genetic resources of honey bees and act as a natural laboratory for studies of adaptation, hybridization, and demographic processes.

## 5. Conclusions

This study increases our knowledge of the complex architecture of the tRNA^leu^-cox2 intergenic region, reporting new haplotypes in Peruvian honey bee populations for the first time. The regions Junín and Piura present a higher frequency of African haplotypes than Lima, and share an Africanized genetic pattern due to human-mediated processes. Lima is characterized by more haplotypes of lineage C due to the frequent importation of queens coming from Chile and artificial selection by professional beekeepers. The scarce presence of the M lineage, still present, could be related to accidental purchases of M ancestry queens or to introductions of European honey bees in 1884. Further studies are needed to characterize honey bees from natural and selected populations on Peruvian territory using the tRNA^leu^-cox2 intergenic region, and to use the information generated in large-scale conservation and improvement programs.

## Figures and Tables

**Figure 1 insects-12-00641-f001:**
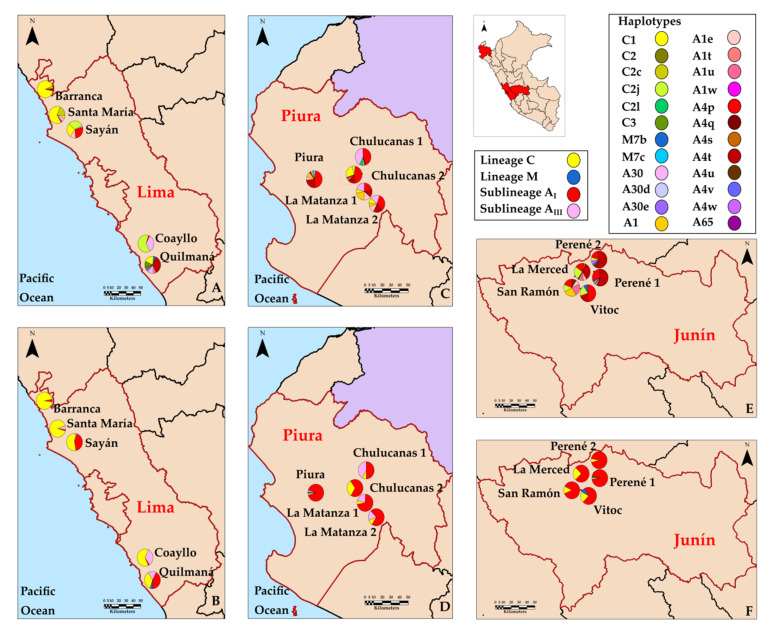
Sampling sites in three regions from Peru. Pie charts display the proportions of haplotypes (**A**,**C**,**E**), and evolutionary lineages and sub-lineages (**B**,**D**,**F**) for Peruvian honey bee populations.

**Figure 2 insects-12-00641-f002:**
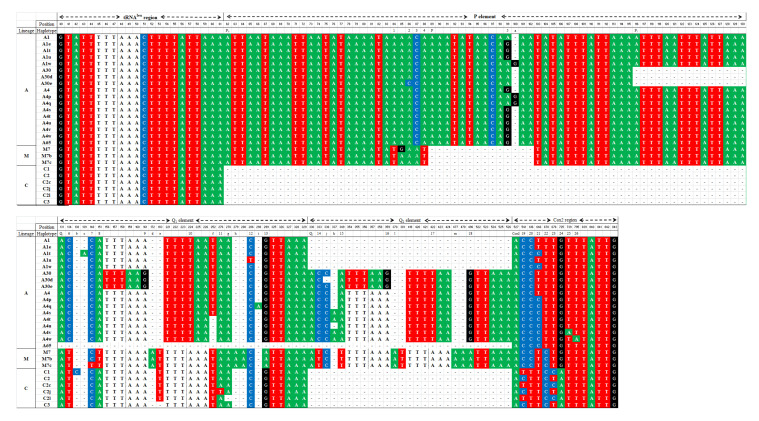
Variable sites of the tRNA^leu^-cox2 intergenic region showing variation among haplotypes of lineages C, A, and M identified in Peruvian honey bee populations. Substitution sites are numbered from 1 to 26. Indels (indicated with a dash) are denoted by letters from “a” to “m”. The large deletions *d* and *d1*, typical of the P and P_1_ forms of the P element, are represented by gray italic letters. *Dra*I recognition sites (TTTAAA) are indicated by nucleotides in bold with a white background. The absence of the P element characterizes lineage C. Sub-lineage A_I_ carries the P_0_ form of the P element, whereas sub-lineage A_III_ is identified by the P_1_ form of the P element (indicated by the “*d1*” deletion). Haplotypes of lineage M are mostly differentiated by the P element (indicated by the “*d*” deletion).

**Figure 3 insects-12-00641-f003:**
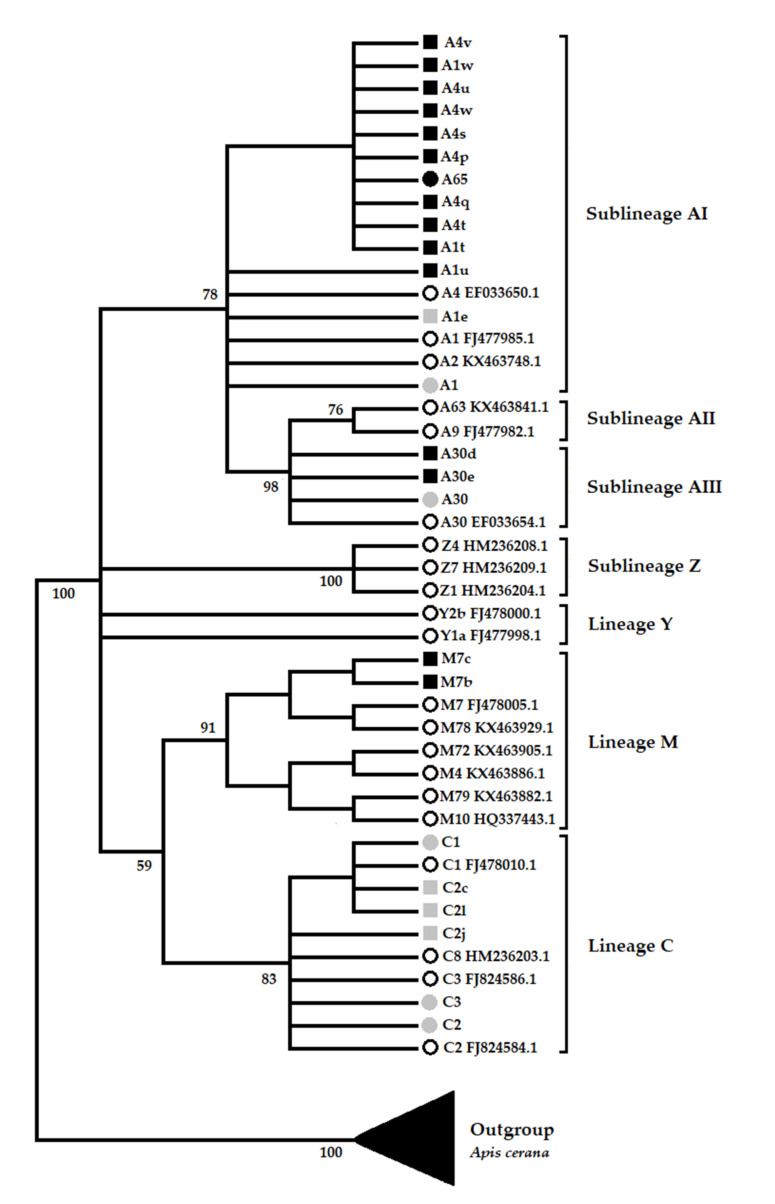
Phylogenetic relationships of haplotypes identified for Peruvian honey bee populations. Sub-lineages A_I_, A_II_, A_III_, and Z belong to the African lineage. Nodes are supported by 1000 bootstraps. Black circles indicate novel haplotypes, light gray circles indicate haplotypes previously reported and found in this study, black squares indicate novel variants, light gray squares indicate variants reported by others and in this study, and white circles indicate reference haplotypes obtained from Genebank.

**Figure 4 insects-12-00641-f004:**
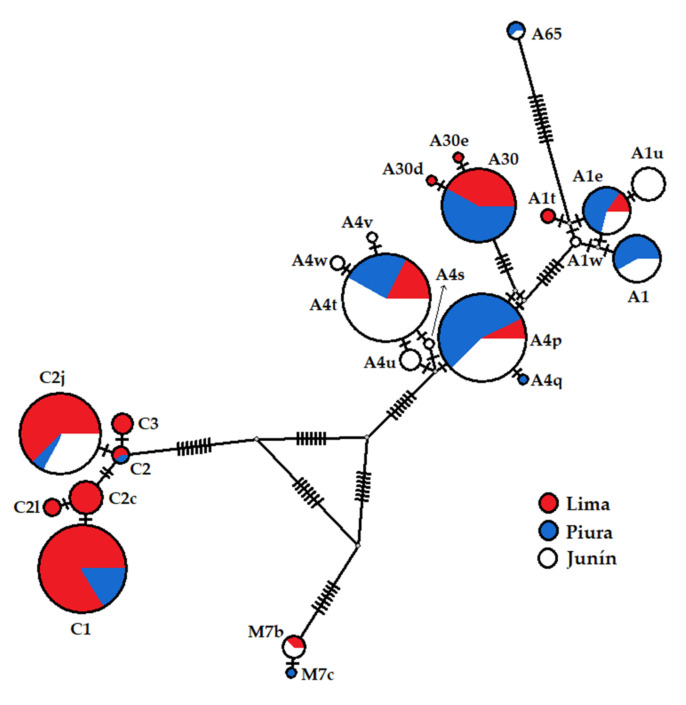
Haplotype network based on tRNAleu-cox2 sequences of *Apis mellifera* from Lima, Piura, and Junín. Hypothetical (unsampled or extinct) haplotypes are indicated as gray-filled circles. The size of circles is proportional to the observed haplotypes frequencies. Colors depict the proportion of individuals identified from the three Peruvian regions sharing the haplotype. Each tick represents a mutational step between haplotypes.

**Figure 5 insects-12-00641-f005:**
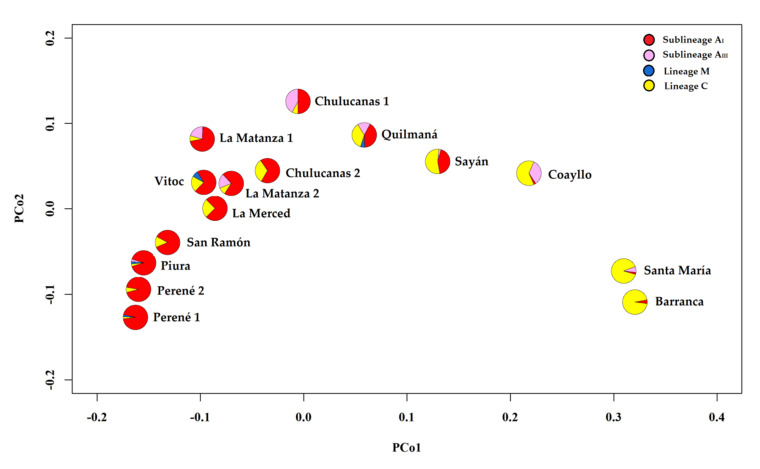
PCoA of sampling sites based on Φ*_PT_* matrix obtained with frequency data for lineages and African sub-lineages. Pie charts display the proportions of lineages and sub-lineages for each sampling sites.

**Table 1 insects-12-00641-t001:** Diversity measures for each of the sampling sites the three Peruvian regions.

Region	Sampling Sites	N	Na	Np	Ne	uh
**Lima**	**Barranca**	42	3	1	1.215	0.181
	**Coayllo**	38	3	1	1.936	0.496
	**Quilmaná**	36	9	1	4.909	0.819
	**Santa María**	39	5	1	2.164	0.552
	**Sayán**	35	5	1	3.840	0.761
**Piura**	**Chulucanas 1**	38	5	1	2.597	0.632
	**Chulucanas 2**	31	6	1	3.302	0.720
	**La Matanza 1**	14	6	1	5.444	0.879
	**La Matanza 2**	52	6	1	4.777	0.806
	**Piura**	31	8	1	3.469	0.735
**Junín**	**La Merced**	32	8	1	4.414	0.798
	**Perené 1**	36	7	1	2.374	0.595
	**Perené 2**	29	6	1	2.572	0.633
	**San Ramón**	35	7	1	5.258	0.834
	**Vitoc**	24	4	1	2.909	0.685

N, number of individuals; Na, mean number of haplotypes; Np, number of private haplotypes; Ne, number of effective haplotypes; uh, unbiased haplotype diversity.

## Data Availability

New haplotypes sequences can be downloaded from Genebank through the accession numbers from MW677198 to MW677221 and additional information from Appendix A.

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
