# Peer review of "Mitochondrial DNA Variation in Peruvian Honey Bee (Apis mellifera L.) Populations Using the tRNAleu-cox2 Intergenic Region"

_insects, 2021, doi:10.3390/insects12070641_

Round 1

Reviewer 1 Report

The tRNAleu-COX2 intergenic region of the mitochondrial DNA has been extensively used for assessing diversity in honey bee populations.  In this study, the authors surveyed mitochondrial DNA variation of Peruvian honey bee populations by using 13 tRNAleu-cox2 intergenic regions and found Lineage A (African) was the dominant (333 colonies, 65.0%), compared with only 173 colonies (204 33.8%) for Lineage C (Eastern European) and six colonies (1.2%) for Lineage M (Western and Northern European). Also, the authors explained the possible reasons and origin of Lineages C and M in Peru. Overall, this paper provides essential information for better understanding the heritage, diversity, relationships among haplotypes, lineages, and genetic patterns in the Peruvian honey bee populations. These results could also help for the honeybee breeding based on genetic background. I would say this is an exciting paper for publication.  

Minor changes:

Line 14: the author wrote: “Only two studies characterized these populations”, could you please more detail about “two studies”, also in lines 76-77, “To date, little information is available on diversity and genetic structure of Peruvian honey bee populations. Only two studies were carried out, to estimate the expansion of Africanized honey bee by using morphometry”.

Line 205-206:  the author wrote: “Within lineage A, the sub-lineage AI (280 colonies, 205 84.1%) was also more ubiquitous (P<0.05) than sub-lineages AIII (53 colonies, 15.9%)”. How do authors calculate the P-value? Please have a paragraph to descript it in material and method.

Lines: 254-255: the author wrote: “Circle size represents is proportional to haplotypes frequencies”. Please correct the grammar “represents is”.

Lines: 353-356: the author wrote: “Taking all these together, the Peruvian territory could still habor important genetic resources of honey bees, in addition it represents a natural laboratory for studies of adaptation, hibridization and demographic processes”. Please correct the grammar “, in addition it”.

Reviewer 2 Report

Dear Authors,

Your manuscript represents interesting scientific work. I have some minor comments that needs to be addressed by authors.

  1. L93-97

What was the principle used in the selection of sampling sites for each Peruvian region?

How were the sampling sites selected?

Why were these five sampling sites chosen in each region?

  1. L96

The Materials indicated that 33 apiaries were examined. It is not entirely clear why there are 33 apiaries?

According to the text of the manuscript:

3 region×5 sampling sites ×(13-15 apiaries) = total, 195-225 apiaries. Where is the error?

  1. L96

The results showed that 512 bee samples were examined (L153). This information should be entered in the Materials and Methods. For example, one individual was examined from each bee colony.

  1. L244-251 (Fig.5)

The results provide interesting data on identifying factors that determine the genetic diversity of honeybees (PCoA).

If I understood correctly, the main axis explains 70.2% of the total genetic variation, and the bee samples compared are clustered according to the evolutionary lineages (A and C). The second axis explained 15.5% of the genetic variation and showed differentiation of sub-lineage AIII samples from to sub-lineage AI samples and lineage C samples.

The authors suggest that the main factor determining genetic diversity is the origin of bees (evolutionary lineages, sub-lineages). If the A lineage samples and C lineage samples are clearly clustered (Coordinate 1), then bee samples of varies sub-lineage are not so  different (Coordinate 2). 

Perhaps, in addition to the genetic component, other factors, such as the presence of different haplotypes in the sample site, the geographical location of apiaries, etc., also influence the determination of the genetic diversity of Peruvian honeybee populations.

At the same time, in the Discussion (L336, 350-351), the authors discuss the role of climatic and environmental factors, present in Peruvian territory, in the formation of the observed mitochondrial diversity in Peru.

  1. Continuing the previous comment.

L338-343

The authors discuss the role of ecological and environmental factors that create selection pressures on the honeybee genome, including mitochondrial DNA.

The authors try to explain the significant diversity of haplotypes by the adaptive evolution of mtDNA.

What are the ecological and geographical conditions in Peru that so many new haplotypes have been identified there?

Why is this not observed in other countries in South America, for example in Colombia?
